# The Association Between the Eosinophilic COPD Phenotype with Overall Survival and Exacerbations in Patients on Long-Term Non-Invasive Ventilation

**DOI:** 10.3390/biom15121728

**Published:** 2025-12-12

**Authors:** Andras Bikov, Balazs Csoma, Andrew Chai, Eleonor Croft, Zsofia Lazar, Andrew Bentley

**Affiliations:** 1Wythenshawe Hospital, Manchester University NHS Foundation Trust, Manchester Academic Health Science Centre, Oxford Road, Manchester M13 9PT, UK; csoma.balazs@semmelweis.hu (B.C.); andrew.bentley@manchester.ac.uk (A.B.); 2Division of Infection, Immunity & Respiratory Medicine, Faculty of Biology, Medicine and Health, The University of Manchester, Manchester M13 9PL, UK; andrew.chai2@nhs.net (A.C.); eleanor.croft@student.manchester.ac.uk (E.C.); 3Department of Pulmonology, Semmelweis University, 1085 Budapest, Hungary; lazar.zsofia@semmelweis.hu

**Keywords:** eosinophil, COPD, exacerbation, hypercapnia, NIV

## Abstract

Background: Long-term non-invasive ventilation (LT-NIV) can prolong life expectancy and may reduce the number of exacerbations in patients with COPD. The eosinophilic phenotype has recently gained significant attention as a treatable trait in COPD. However, it is less known how this phenotype relates to exacerbations and mortality in patients who are set up on LT-NIV. Methods: A total of 191 patients with COPD (65 ± 8 years, 55% women) who were setup on LT-NIV and followed-up (28/15–49/months) at our tertiary centre were analysed. The eosinophilic phenotype was defined by using an accepted cutoff for blood eosinophil count (≥300 cells/µL). Results: A total of 37 patients had the eosinophilic phenotype (66 ± 9 years, 60% women). There was a higher reduction in the number of exacerbations (1.0/−1.0–3.2/ vs. 0.05/−1.4–1.63/, *p* < 0.01) and a trend for a reduction in the rate of hospitalisations (1.0/−1.0–2.0/ vs. 0.0/0.0–1.0/, *p* = 0.07) post-NIV setup in the eosinophilic group. Most importantly, patients with high eosinophil counts had longer overall survival (34/15–74/ vs. 28/15–47/ months, *p* = 0.02, adjusted for covariates). Conclusions: The eosinophilic COPD phenotype seems to show better clinical responses to long-term NIV than patients without this trait. Further mechanistic studies are warranted to analyse this association.

## 1. Introduction

Chronic obstructive pulmonary disease (COPD) is a common disorder that is characterised by chronic symptoms and airflow limitation, usually caused by exposure to noxious particles. The airflow limitation is usually progressive on the background of chronic inflammatory changes [1]. With worsening lung function, patients with COPD are at increased risk for the development of chronic respiratory failure, characterised by chronic hypoxia and hypercapnia [2]. Chronic hypercapnia can be detrimental in COPD by contributing to the development of pulmonary hypertension [3] and reduced antimicrobial defence [4]. Consequently, hypercapnia is an independent risk factor for mortality in patients with COPD [5]. Long-term non-invasive ventilation (LT-NIV) is an effective treatment to improve or normalise chronic hypercapnic respiratory failure in patients with COPD [6]. It prolongs overall survival [7,8,9], and it may reduce the number of exacerbations [7] in patients with hypercapnic COPD. However, it is not clear which patients would benefit the most from this intervention. Biomarkers that predict treatment responsiveness could be helpful in selecting the most appropriate patients, allowing more effective resource utilisation.

Chronic airway inflammation in COPD is heterogenous and is usually characterised by increased numbers of macrophages and neutrophils [1]. However, around one-third of patients with COPD have airway eosinophilia [10], which is associated with a distinct clinical phenotype [11]. Airway eosinophilia is difficult to investigate directly in clinical practice due to limited access to induced sputum or bronchoalveolar lavage. On the one hand, these techniques are not fully safe, and, on the other hand, patients may not be able to produce adequate-quality sputum [11]. Therefore, the blood eosinophil count (BEC) is often used to assess the eosinophilic endotype. Various blood eosinophil count thresholds were tested, and ≥300 cells/µL was found to be an optimal cutoff for detecting sputum eosinophilia [12]. This cutoff was also found to be predictive of safe inhaled corticosteroid withdrawal in patients with COPD [13]. Increased blood eosinophils are predictive of steroid responsiveness [14] and are associated with exacerbation frequency, especially in patients that are steroid-naïve [15,16]. However, data on blood eosinophils, exacerbation risk, and mortality in COPD are contradictory [17,18,19,20,21].

On the one hand, eosinophils have an important antimicrobial role in the airways [10] and may also balance neutrophilic inflammation [22]. On the other hand, whilst hypercapnia reduces the antimicrobial potential of macrophages [4], it increases interleukin-4 (IL-4) levels, and IL-4 is a key cytokine stimulating eosinophils [23]. Therefore, it is plausible that the interaction between chronic hypercapnia and airway eosinophilia in patients with COPD is associated with an altered clinical response. However, the eosinophilic phenotype has not been investigated in the relation of LT-NIV in COPD. Considering that patients with hypercapnic COPD may have unique airway inflammation [6], data on eosinophilic COPD may not be directly translatable to this population. Therefore, the aim of this project was to assess whether patients with the eosinophilic phenotype have different benefits from long-term NIV.

## 2. Materials and Methods

### 2.1. Design and Subjects

We analysed patients with COPD who participated in our service evaluation project, which aimed to assess the effect of LT-NIV in patients with COPD who were set up on LT-NIV between September 2011 and August 2023 at our tertiary centre. Out of the 392 patients registered in this project, we excluded patients who were set up on LT-NIV during their acute admission with respiratory failure (*n* = 151) and patients who did not have full blood count results at LT-NIV setup (*n* = 50). All participants in this analysis were set up in their stable state (at least 4 weeks after their exacerbation), according to the practices suggested by the HOT-HMV study [7].

COPD was initially diagnosed based on symptoms, suggestive medical history and lung function test. The diagnosis was validated by the clinical team. Data on comorbidities and medications were based on patient reports and electronic patient records, and the Charlson Comorbidity Index (CCI) [24] was calculated. Symptoms and frailty were assessed based on the modified Medical Research Council (mMRC) questionnaire and the Clinical Frailty Score (CFS). Exacerbations and hospital admission 12 months before LT-NIV setup and during the follow up period post-setup were collected from the hospital records, the Greater Manchester Care Records and patients’ reports. The exacerbation was considered moderate if it required systemic corticosteroids and/or antibiotics and severe if it required hospital admission. If the patient needed repeated courses or corticosteroids or antibiotics within 30 days, we considered this a single event. If the exacerbation required treatment with antibiotics, it was labelled as “infective”. The decision to prescribe steroids or antibiotics was made by the primary care providers or, in case of a hospital admission, by the primary care team. Due to the variable follow-up time, exacerbations and hospital admissions were annualised after LT-NIV.

Following a review by a senior clinician, patients with chronic hypercapnia (defined by pCO_2_ ≥ 6 kPa and normal pH) attended the North West Ventilation Unit for inpatient LT-NIV setup. As part of routine clinical practice, patients had blood tests, including full blood count, renal function and electrolytes, capillary blood gas tests, electrocardiograms and chest X-rays before LT-NIV setup. Nocturnal oximetry was performed in 90 patients. We adjusted LT-NIV intensity (pressure support, respiratory rate and hours of use) based on blood gases, supported by nocturnal oximetry, with the aim of reducing pCO_2_ to achieve normocapnia or significant reduction, similar to other practices [7,8]. Following their discharge, the patients had routine follow-up appointments at 3, 6 or 12 months depending on their clinical need. During the follow-ups, the medical management was optimised, the LT-NIV settings and the interfaces were adjusted based on the blood gas results, and data downloads from the devices.

Blood collection, obtaining clinical information, and the intervention (LT-NIV) were part of the routine clinical service. Patients were not randomised into study groups, and this project was not aimed at generalising information. Following assessment using the Medical Research Council decision tool (https://www.hra-decisiontools.org.uk/research/, accessed on 1 September 2022), as later confirmed by the Health Research Authority (on 1 May 2024), this project was not considered to be research. Therefore, this project was registered (registration number HL075) and approved (on 16 October 2022) as a service evaluation by the Trust Clinical Governance body. Data collection and analysis adhered to the Caldicott Principles, and data were not shared outside the clinical team. Service evaluation projects, when conducted, are part of the Trust’s Public Task in order to inform evidence-based practice and improve patient care, relying on Articles 6.1(e) and 9.2(h). The Trust does not rely on explicit consent when performing service evaluation studies; therefore, no written consent was obtained.

### 2.2. Statistical Analysis

JASP 0.14 (JASP Team, University of Amsterdam, Amsterdam, the Netherlands) and Statistica 12 (StatSoft, Inc., Tulsa, OK, USA) were used for statistical analysis. Patients were divided into high (≥300 cells/µL) and low (<300 cells/µL) eosinophil groups. The demographics and clinical characteristics between the low- and high eosinophil groups were compared with Student’s, Mann–Whitney and Chi-square tests. The variations in BEC at LT-NIV setup were assessed with the Wilcoxon test. Overall survival, exacerbation-free survival, hospitalisation-free survival, and infective exacerbation-free survival were compared with log-rank tests between the high and low eosinophil groups. These were adjusted for age, body mass index (BMI), forced expiratory volume in one second (FEV_1_, %predicted), CCI, mMRC, the average hours of NIV use and the number of exacerbations the year before LT-NIV setup using Cox regression. Sensitivity analyses were conducted treating BEC as a continuous variable and using BEC ≥ 100 cell/µL or BEC ≥ 200 cell/µL as a cutoff for the high blood eosinophil group. Further sensitivity analyses in patients taking inhaled corticosteroids/long-acting β-agonist/long-acting muscarinic antagonists (ICS/LABA/LAMA) were performed. Changes in exacerbations and hospital admissions following LT-NIV setup were compared between the two groups with Mann–Whitney tests as well as non-parametric ANCOVA tests adjusted for age, gender, FEV_1_ (% predicted), hours of NIV use and the number of exacerbations the year before LT-NIV setup. Data are expressed as mean ± standard deviation or median/interquartile range/. A *p* value < 0.05 was considered significant.

As this was a retrospective analysis of data collected in a service evaluation project, no formal power calculations were performed.

## 3. Results

### 3.1. Comparison of the High Eosinophil and Low Eosinophil Groups

Patients with high BEC had higher numbers of exacerbations and hospitalisations, more severe airflow obstruction and overnight hypoxia (all *p* < 0.05). There was no difference between the two groups in age, gender, BMI, the symptom scores, COPD medications, or capillary blood gas results; however, there was a tendency toward increased prescription of ICS/LABA/LAMAs in the high eosinophil group (all *p* > 0.05, Table 1).

There was no difference in the prevalence of comorbidities or medications that can contribute to hypoventilation between the two groups. However, the prevalence of asthma tended to be higher and the prevalence of allergic rhinitis tended to be lower in the high eosinophil group (Table 2).

There was no difference between the two groups in the intensity of long-term NIV. The proportion of patients in whom acceptable pCO_2_ (defined by pCO_2_ <7 kPa) and normocapnia (defined by pCO_2_ <6 kPa) were achieved was also similar at the time of LT-NIV setup (all *p* > 0.05, Table 3).

### 3.2. Stability of BEC at LT-NIV Setup

Forty-eight patients had full blood count twice during their elective admission. In these patients, we noticed a significant change in the eosinophil counts (from 120/6–220/ to 200/130–310/ G/L, *p* < 0.01). In 70% of cases blood eosinophils stayed below 300 cells/µL and in 11% stayed above 300 cells/µL on both tests. In the remaining 19% they either increased (17%) or decreased (2%, Figure 1).

### 3.3. Overall Survival, Exacerbation-Free Survival and Hospitalisation-Free Survival in Patients with and Without the Eosinophilic Phenotype

The median follow up was 28/15–47/months in the low eosinophil group and 34/15–74/months in the high eosinophil group. The percentages of active participants in each group during the follow-up period are summarised in Table 4. During the follow up period, the average hours of LT-NIV use were similar between the high (7.1 ± 4.0 h) and low eosinophil groups (7.2 ± 3.7 h, *p* = 0.95).

Patients with the eosinophilic phenotype tended to have a prolonged overall survival (*p* = 0.06). However, there was no difference between the two groups in exacerbation-free survival (*p* = 0.69), hospital admission-free survival (*p* = 0.90) or infective exacerbation-free survival (*p* = 0.68, Figure 2).

However, when the analyses were adjusted for age, BMI, FEV_1_, CCI, mMRC, the average hours of NIV use, and the number of exacerbations the year before LT-NIV setup, the high eosinophil group was significantly related to overall survival (*p* = 0.02), but not the exacerbation-free survival (*p* = 0.67), hospital admission-free survival (*p* = 0.64) or infective exacerbation-free survival (*p* = 0.80). The risk ratio (95% confidence interval) for mortality in the eosinophilic group was 0.41 (0.20–0.87).

The results were similar when only patients who took ICS/LABA/LAMAs were analysed (*p* = 0.04, *p* = 0.99, *p* = 0.67, and *p* = 0.77 for overall survival, exacerbation-free survival, hospital admission-free survival, and infective exacerbation-free survival, respectively).

When investigating BEC as a continuous variable, no association was found with the overall survival (*p* = 0.60). Similarly, when using BEC ≥ 100 cells/µL or BEC ≥ 200 cells/µL as cutoffs, the relationships with overall survival were not significant (*p* = 0.60 and *p* = 0.51).

### 3.4. Changes in Exacerbations and Hospitalisations in Patients with High and Low BEC

Following LT-NIV setup patients in the high eosinophil group tended to experience higher reduction in exacerbations (1.0/−1.0–3.2/ vs. 0.05/−1.4–1.63/, *p* = 0.06) and hospital admissions (1.0/−1.0–2.0/ vs. 0.0/0.0–1.0/, *p* = 0.06). In contrast, there was no difference in the changes in infective exacerbations (0.0/−1.2–2.0/ vs. 0.0/−1.2–1.4/, *p* = 0.31, Figure 3). However, after adjustment, the changes in exacerbations became significant (*p* < 0.01) while changes in hospital admissions (*p* = 0.07) and infective exacerbations (*p* = 0.46) remained insignificant.

## 4. Discussion

In this study we investigated whether the eosinophilic COPD phenotype is associated with different benefits from LT-NIV. We found that patients with COPD and BEC > 300 cells/µL at the initiation of ventilation had prolonged survival and greater reductions in exacerbations.

It is worth noting that patients with the eosinophilic phenotype had more severe and more active disease before LT-NIV setup. Similar to the results published in a large cohort study [25], patients with higher BEC had worse FEV_1_. This could be explained by both mechanistic studies describing the role of eosinophils in airway remodelling [10] and the fact that patients with eosinophilic COPD have more rapid lung function decline [26]. A lower FEV_1_ is a strong predictor for both exacerbations and mortality in COPD [27]. Furthermore, in line with previous studies, the numbers of exacerbations as well as hospitalisations were higher in the eosinophilic group [15,16]. However, the relationship between past exacerbations and eosinophils is more obvious in patients who are ICS-naïve [16]. Therefore, our results need to be interpreted carefully as most patients in our cohort took ICSs. Nevertheless, exacerbations are predictive of further events [28] and are associated with reduced overall survival [29]. However, despite having more advanced disease and a higher burden of exacerbations, patients with a higher BEC had favourable survival and exacerbation reduction after LT-NIV. Of note, all analyses were adjusted for FEV_1_ and exacerbations in the year before LT-NIV setup. Importantly, blood gas parameters, BMI and the severity of comorbid obstructive sleep apnoea pre-setup were similar in the two groups. Not surprisingly, NIV prescription, the successful achievement of acceptable pCO_2_ levels, and, importantly, adherence to NIV post-setup were also not different. Importantly, the main analyses were adjusted for the hours of NIV use. Therefore, differences in post-NIV survival and exacerbation rate were not due to variances in LT-NIV effectiveness.

The most important finding of our study is that patients with high BECs had prolonged overall survival. However, the relationship between BEC and overall survival was not linear and was significant only in patients with BEC > 300 cells/µL. Interestingly, the results are similar to those found in other COPD cohorts [18,19,21] but not to those of all studies [20]. Furthermore, the BEC thresholds for survival benefits were also variable [18,19,21]. This highlights that unexplored confounding factors, such as the various COPD phenotypes or airway microbiome, might play a role. Most importantly, the mortality analyses were adjusted for age, BMI, FEV_1_, symptom scores, exacerbation rate, and the Charlson Comorbidity Index, factors that have been previously found to predict mortality in COPD [27,30,31,32]. There was no difference in the prevalence of relevant comorbidities, and medications that can depress respiratory drive were also similar between the two groups [32]. Unfortunately, due to the nature of this study, whilst we were able to assess the occurrence of death, causes of mortality were not available. Therefore, the mechanism needs to be investigated in separate studies. Importantly, the exact cause of death between the high and low BEC groups could also reveal specific pathological pathways affected by eosinophils. Nevertheless, a potential explanation could be the susceptibility to pneumonia in patients with low eosinophil counts [33]. Furthermore, eosinopenia has been associated with worse outcomes in patients hospitalised with COPD exacerbations [34]. Patients with >2% blood eosinophils who required admission to intensive care and either NIV or invasive mechanical ventilation due to COPD exacerbation had better survival compared to those with ≤2% blood eosinophils [35].

We also found a higher reduction in exacerbations after LT-NIV setup in the eosinophilic group. It is worth acknowledging that the number of exacerbations before LT-NIV was already higher in patients with blood eosinophilia; therefore, numerical reduction in the exacerbations could have been due to imbalances between the groups. To avoid this bias, the analyses were adjusted for the baseline variables. Beyond methodological factors, the observed findings can be explained by immunological factors. First, higher levels of type 1 inflammatory mediators were associated with higher susceptibility to exacerbations and mortality [17]. Patients with eosinophilic COPD have lower levels of type 1 mediators in the airways [22]. In addition, eosinophils have a protective effect against both bacterial and viral infections in COPD [10]. This leads to an inverse relationship between eosinophil counts and bacterial load in the airways [36]. Second, hypercapnia increases IL-4 levels in vivo and in vitro [23]. This suggests that patients with hypercapnia are more likely to have airway eosinophilia, which is associated with non-infective exacerbations. In line with this, reduction in hypercapnia via LT-NIV can therefore lower the number of non-infective exacerbations, leading to an overall reduction in exacerbations. In addition, non-infective eosinophilic exacerbations may better respond to systemic corticosteroids [11]. This hypothesis was supported by the lack of difference in the changes in the infective exacerbations between the two groups. Whilst the hypothesis is plausible, this needs to be investigated in mechanistic studies.

It is noteworthy that BEC showed significant short- and long-term variations in COPD cohorts [18,37,38]. Similar to two large, population-based studies, only 11% of our patients had persistently high eosinophil numbers [18]. However, our results need to be interpreted carefully, as repeated blood tests during a stable state were performed in only a subgroup of patients. The acute effect of NIV setup on variations cannot be excluded either. We used the first blood test when categorising patients into the two groups. Sensitivity analyses in patients with permanent eosinophilia as well as in those with increasing and decreasing trends are warranted. However, the sample size precluded such analyses. Furthermore, blood eosinophilia can occur in multiple disorders, including allergic diseases, dermatological disorders, gastrointestinal disorders, rheumatological diseases, vasculitides, neoplasms, as well as parasitic and fungal infections [39]. These disorders were not specifically investigated in our cohort. Therefore, the results need to be interpreted with caution. Eosinophils may increase during exacerbations in patients with otherwise normal blood eosinophil counts during the stable state [40]. Furthermore, only around half of eosinophilic exacerbations are followed by another eosinophilic exacerbation [41]. To avoid this bias, we excluded patients who were set up on LT-NIV during their acute deterioration. Nevertheless, the results on the predictive value of blood eosinophils taken during COPD exacerbations on long-term outcomes are contradictory [41,42,43].

Patients on ICS/LABA/LAMAs experience prolonged survival compared to those on LABA/LAMAs [44,45]. Although the magnitude of exacerbation reduction is associated with higher BEC [46], the mortality reduction was not investigated in relation to BEC [47]. To mitigate this confounding factor, survival analyses were performed in patients on ICS/LABA/LAMAs. These analyses confirmed the results in the overall population. The prescription of fix and open triple combinations was variable in the cohort, and, due to the low number of subjects, no subgroup analyses were performed. It is possible that during the follow-up period patients became more adherent to their inhaler therapy. For those with high BEC, this could have translated into better clinical outcomes [46]. Whilst this hypothesis is plausible, it could not be tested in a service evaluation project.

This study has some limitations. First, BEC data were available only at baseline, and we did not analyse longitudinal changes. We also did not have access to historical BEC results. Second, during the follow-up appointments, both LT-NIV and medications were optimised. However, the analyses were not adjusted for these changes. Although there was no difference between the two groups in comorbidities, COPD medications or NIV prescriptions, the potential bias cannot be fully excluded. Third, the emphysematous COPD phenotype has previously been related to reduced survival in patients with COPD [21], including those on LT-NIV [48]. However, detailed lung function tests, including lung volumes and diffusion capacity, were not available for many patients; therefore, this phenotype was not analysed. Fourth, we did not have data on airway microbiology. Specific bacteria, including Haemophilus, Moraxella, and Pseudomonas, are more prevalent in patients with higher exacerbation burden [49], and it is possible that these bacteria were more prevalent in the low eosinophilic group [36]. Fifth, exacerbations, including infective exacerbations, were defined by prescriptions initiated by external providers and therefore were prone to bias. Sixth, the sample size did not allow investigating specific subgroups, such as those with variable or persistently high blood eosinophils. Finally, this was a service evaluation project, which was not designed to produce generalisable findings. We used clinical information that was obtained during routine clinical practice, and service evaluations do not allow requesting tests for research purposes. As such, data that would allow phenotyping our patients in more detail were not available. Further, interventional multi-centre studies are warranted to test our hypothesis. Service evaluations do not allow us to study causality; therefore, mechanistic studies should focus in explaining our findings. We believe that our results will provide the basis for designing such trials.

## 5. Conclusions

In summary, the eosinophilic phenotype (defined by blood eosinophils ≥300 cells/µL) may be associated with favourable outcomes in patients with COPD starting long-term NIV. Specifically, patients with higher BECs experience a survival benefit and a larger reduction in their exacerbation rate following LT-NIV setup. However, the results need validation by independent studies.

## Figures and Tables

**Figure 1 biomolecules-15-01728-f001:**
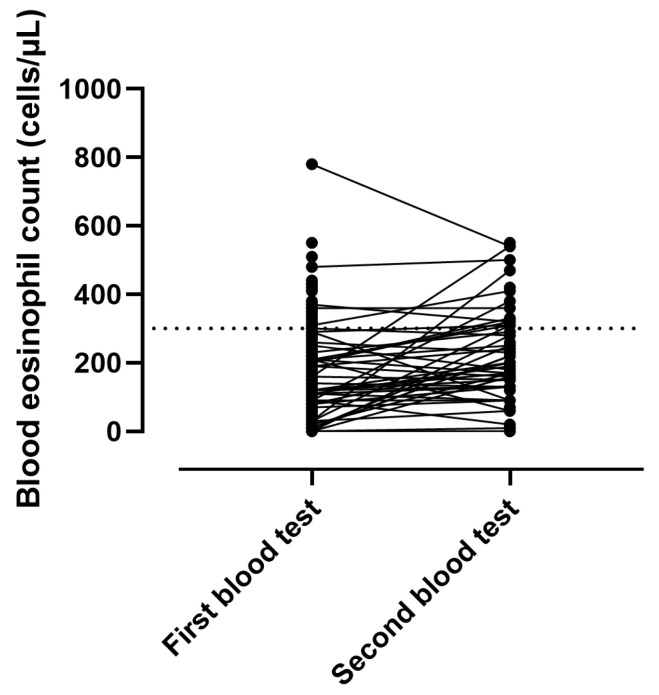
Stability of blood eosinophil counts during LT-NIV setup. Dashed line represents 300 cells/µL.

**Figure 2 biomolecules-15-01728-f002:**
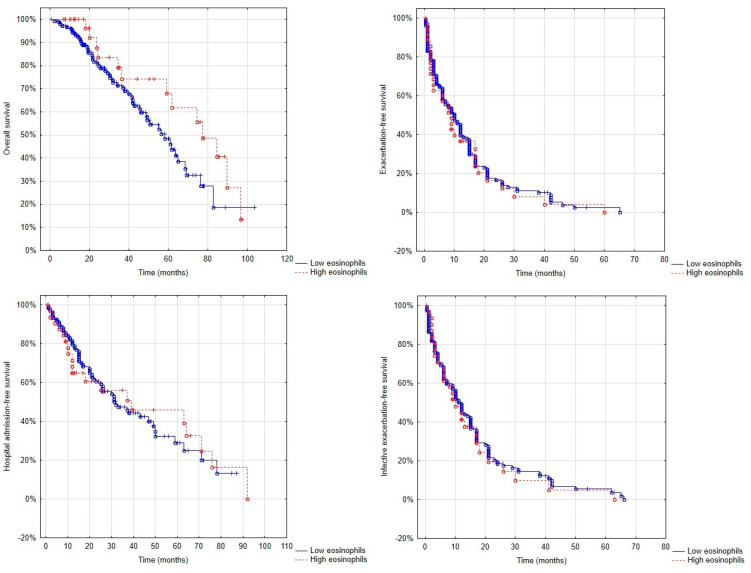
Survival analyses post LT-NIV setup. Kaplan–Meier curves for overall survival (**upper left**), exacerbation-free survival (**upper right**), hospital admission-free survival (**lower left**), and infective exacerbation-free survival (**lower right**). Assessed by adjusted Cox regression analyses, in the high eosinophil group, the risk ratios (95% confidence interval) were 0.41 (0.20–0.87), 0.91 (0.59–1.40), 0.87 (0.48–1.57), and 0.94 (0.59–1.50) for overall survival, exacerbation-free survival, hospital admission-free survival, and infective exacerbation-free survival, respectively.

**Figure 3 biomolecules-15-01728-f003:**
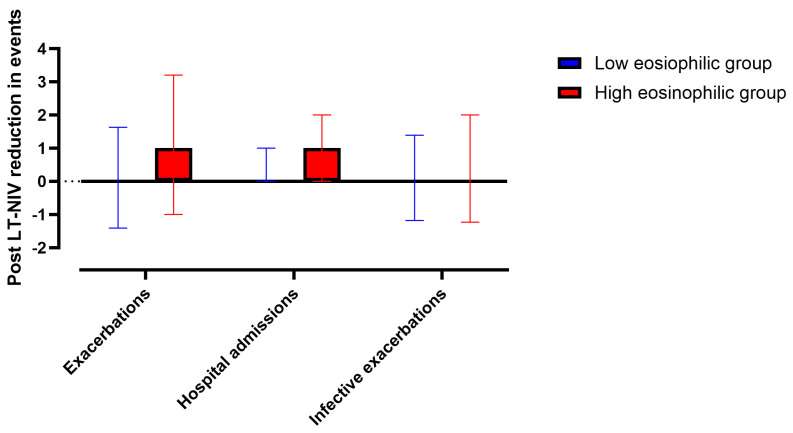
Reduction in exacerbations, hospital admissions and infective exacerbations post-LT-NIV setup. Post LT-NIV reduction was calculated as the difference between the number of events in 12 months before LT-NIV setup and the annualised number of events after LT-NIV. Data are expressed as median with interquartile range.

**Table 1 biomolecules-15-01728-t001:** Comparison of the low and high eosinophil groups before LT-NIV setup.

	Low Eosinophil Group (*n* = 154)	High Eosinophil Group (*n* = 37)	*p* Value
Age (years)	65.7 ± 9.3	64.4 ± 8.5	0.43
Gender (female, %)	60	57	0.69
BMI (kg/m^2^)	31.7 ± 10.3	32.5 ± 9.5	0.67
BMI > 30 kg/m^2^ (%)	57	56	0.94
BMI > 35 kg/m^2^ (%)	38	35	0.81
Number of exacerbations in the year before setup	2/1–4/	4/2–8/	0.01
Frequent exacerbations (≥2 exacerbations/year, %)	73	79	0.47
Number of hospital admissions in the year before setup	1/0–2/	1/1–2/	0.04
Number of infective exacerbations in the year before setup	2/1–3/	3/1–5/	0.03
Never/ex-/current smokers (%)	1/30/69	3/29/68	0.80
Cigarette pack years	48.4 ± 24.9	44.0 ± 19.0	0.36
mMRC	3.7 ± 0.7	3.7 ± 0.7	0.96
CFS	4.9 ± 1.0	4.9 ± 1.2	0.94
On ICS (%)	83	92	0.17
ICS dose (μg budesonide equivalent)	800/200–800/	800/250–800/	0.52
On LABAs (%)	91	97	0.22
On LAMAs (%)	83	92	0.17
On ICS/LABAs (%)	11	6	0.34
On LABA/LAMAs (%)	10	6	0.40
On ICS/LABA/LAMAs (%)	72	86	0.06
Not on fix triple combination (%)	17	32	
On fix BP/FF/GB (%)	31	30	
On fix FF/VT/UB (%)	23	24	
On fix B/FF/GB (%)	1	0	
On systemic corticosteroids (%)	7	8	0.74
BEC (cells/µL)	140/90–200/	370/320–425/	<0.01
Capillary blood pH	7.42 ± 0.06	7.41 ± 0.05	0.34
Capillary blood pCO_2_ (kPa)	7.46 ± 1.37	7.67 ± 1.05	0.40
Capillary blood pO_2_ (kPa)	7.79 ± 1.24	7.63 ± 0.86	0.49
HCO_3_^−^ (mmol/L)	32.4 ± 4.17	32.7 ± 4.90	0.75
FEV_1_ (L)	1.00 ± 0.41	0.84 ± 0.32	0.05
FEV_1_ (% predicted)	41.2 ± 15.8	31.0 ± 15.2	<0.01
FVC (L)	2.15 ± 2.06	1.99 ± 0.65	0.32
FVC (% predicted)	71.5 ± 19.5	67.9 ± 22.3	0.55
ODI (events/hour)	21.2 ± 26.4	20.1 ± 20.0	0.89
T90 (%)	69.2 ± 36.1	90.3 ± 15.6	0.04

BEC—blood eosinophil count, B/FF/GB—budesonide/fluticasone furoate/glycopyrronium bromide, BP/FF/GB—beclomethasone diproprionate/fluticasone furoate/glycopyrronium bromide, BMI—body mass index, mMRC—modified Medical Research Council questionnaire, CFS—Clinical Frailty Score, ICSs—inhaled corticosteroids, LABAs—long-acting β-agonists, LAMAs—long-acting muscarinic antagonists, FEV_1_—forced expiratory volume in 1 s, FF/VT/UB—fluticasone furoate/vilanterol trifenatate/umeclidium bromide, FVC—forced vital capacity, ODI—oxygen desaturation index, T90—percentage of time spent with oxygen saturation below 90%.

**Table 2 biomolecules-15-01728-t002:** Comparison of comorbidities and medications between the two groups.

	Low Eosinophil Group (*n* = 154)	High Eosinophil Group (*n* = 37)	*p* Value
CCI	2.2 ± 1.5	2.5 ± 1.6	0.22
Chronic heart failure (%)	29.3	30.3	0.91
Ischaemic heart disease (%)	25.2	24.2	0.91
Cerebrovascular disease (%)	8.6	8.8	0.97
Pulmonary hypertension (%)	12.7	18.2	0.40
Type 2 diabetes (%)	31.1	44.1	0.15
Chronic kidney disease (%)	10.7	12.1	0.81
Asthma (%)	20.0	33.3	0.09
Bronchiectasis (%)	24.0	35.3	0.18
Obstructive sleep apnoea (%)	26.5	23.5	0.72
Depression (%)	32.0	33.3	0.88
Anxiety (%)	16.6	21.2	0.52
Kyphoscoliosis (%)	1.3	5.9	0.10
Allergic rhinitis (%)	7.1	0.0	0.09
Eczema (%)	1.9	0.0	0.16
Allergic bronchopulmonary aspergillosis (%)	0.6	0.0	0.62
On opioids (%)	46.8	45.9	0.93
On benzodiazepines (%)	18.8	16.2	0.71

CCI—Charlson Comorbidity Index.

**Table 3 biomolecules-15-01728-t003:** Comparison of LT-NIV parameters and effectiveness between the two groups.

	Low Eosinophil Group (*n* = 154)	High Eosinophil Group (*n* = 37)	*p* Value
IPAP (cmH_2_O)	24.0 ± 4.5	25.2 ± 3.9	0.16
EPAP (cmH_2_O)	6.3 ± 2.2	5.9 ± 1.8	0.38
PS (cmH_2_O)	17.7 ± 4.4	18.7 ± 4.0	0.21
Back-up rate (breath/min)	14.4 ± 1.2	14.2 ± 1.1	0.36
Prescribed hours of use	9.5 ± 2.4	9.9 ± 3.5	0.40
pCO_2_ < 7 kPa achieved during LT-NIV setup (%)	79	75	0.68
Normocapnia achieved during LT-NIV setup (%)	36	29	0.44

EPAP—expiratory positive airway pressure, IPAP—inspiratory positive airway pressure, PS—pressure support, LT-NIV—long-term non-invasive ventilation.

**Table 4 biomolecules-15-01728-t004:** Percentage of patients on active follow-up.

	10 months	20 months	30 months	40 months	50 months	60 months	70 Months	80 Months	90 Months	100 months
Low eosinophil group	89	61	47	34	21	14	7	2	1	1
High eosinophil group	89	59	51	41	38	30	27	16	5	0
*p* value	0.97	0.86	0.62	0.49	0.04	0.03	<0.01	<0.01	0.04	0.62

## Data Availability

Anonymised data are available from the corresponding author upon reasonable request.

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
