# Peer review of "The Association Between the Eosinophilic COPD Phenotype with Overall Survival and Exacerbations in Patients on Long-Term Non-Invasive Ventilation"

_biomolecules, 2025, doi:10.3390/biom15121728_

Round 1

Reviewer 1 Report

Comments and Suggestions for Authors

The manuscript ref biomolecules-3980483 « The eosinophilic COPD phenotype is associated with longer 2 overall survival and greater reduction in exacerbations in patients on long-term non-invasive ventilation » by Bikov et al is a well-written and interesting manuscript reporting overall survival and exacerbation-free survival in COPD subjects treated with LT-NIV, depending on blood eosinophil count at inclusion.

The topic is original, however, the methodology suffers from several points that would need clarification.

Major points :

  1. COPD is a vastly heterogenous disease, including features of bronchial disease, and features of emphysema, which is not adressed in the manuscript. Are the two groups really comparable ? Data regarding asthma history, atopy, or ABPA, ENT diseases, CT-scan bronchiectasis and emphysema should be provided.
  2. Overall survival can be impacted notably by cardio-vascular diseases, frequent in severe COPD. Details on comorbidities should be provided in addition to the Charlson index, including cardiovascular diseases, obesity categories (BMI>30, >35), sleep apnea, as well as drugs potentially inducing hypoventilation
  3. Airway microbiology data should be provided, both at exacerbation in the year before inclusion, and during the follow-up (microbial colonisation ? Pseudomonas ? other pathogen associated with overall survival and exacerbation risk ?)
  4. The classification of the patients in the high/low BEC groups is questionable. When first assessed for the BEC, were the patients off systemic corticosteroids for at least 4-6 weeks? The authors describe variations in BEC in a subgroup of patients with consecutive measurements, with 17% of the patients in the low BEC group having BEC >300 in the later assessment. How were those patients finally classified? I suggest either considering those patients with elevated BEC at exacerbation as « T2high » (similarly to severe asthma), or excluding them. I also suggest that BEC in the two years before enrollment is considered to re-categorise the subjects accordingly.
  5. The mean duration of follow-up is not provided. Given the date of LT-NIV between 2011 and 2023, it should be highly heterogeneous between patients. I therefore wonder if the survival curves indicating « 100 months follow-up » in the legend of the x-axis is correct. The number of patients followed up for 20, 40, … months should at least appear under the x-axis.

Minor points

  1. The title should does not totally reflect the results and should be toned-down
  2. Table 1 :
    1. BEC ranges should be provided for both groups
    2. AHR could be added

Author Response

Comment: COPD is a vastly heterogenous disease, including features of bronchial disease, and features of emphysema, which is not addressed in the manuscript. Are the two groups really comparable ? Data regarding asthma history, atopy, or ABPA, ENT diseases, CT-scan bronchiectasis and emphysema should be provided.

Response: We fully agree that bronchitis and emphysema should be considered when interpreting our findings, as the latter has been associated with reduced survival in patients with COPD on LT-NIV (Budweiser et al., Chest, 2007). Unfortunately, many patients did not have detailed lung function tests, including lung volumes and diffusion capacity; therefore, this confounder has not been investigated. We have expanded the limitations in the manuscript. Please, see Discussion, last paragraph.

Skin prick tests or specific IgE are not routinely performed in COPD in United Kingdom; therefore, the data on atopy were not available. This has been acknowledged in the original manuscript.

We added data on the prevalence of asthma, allergic rhinitis and bronchiectasis to the manuscript. The prevalence of comorbid asthma tended to be higher in the eosinophilic group, but there was no difference in the other comorbidities. See Table 2.

Comment: Overall survival can be impacted notably by cardio-vascular diseases, frequent in severe COPD. Details on comorbidities should be provided in addition to the Charlson index, including cardiovascular diseases, obesity categories (BMI>30, >35), sleep apnea, as well as drugs potentially inducing hypoventilation

Response: Thank you very much. We have added the requested information to the manuscript. See Tables 1 and 2. There was no difference in the prevalence of comorbidities or medications potentially inducing hypoventilation. We have also expanded the Discussion. See 3rd paragraph.

Comment: Airway microbiology data should be provided, both at exacerbation in the year before inclusion, and during the follow-up (microbial colonisation ? Pseudomonas ? other pathogen associated with overall survival and exacerbation risk ?)

Response: We agree the airway microbiology is an important factor for exacerbation risk and mortality in COPD. However, patients were referred to our tertiary centre from various external hospitals and sputum microbiology data is scarcely available for these patients. We used clinical information that was gained during routine clinical practice, and service evaluations do not allow requesting additional tests for the study. We have expanded the limitations. Please, see Discussion, last paragraph.

Comment: The classification of the patients in the high/low BEC groups is questionable. When first assessed for the BEC, were the patients off systemic corticosteroids for at least 4-6 weeks? The authors describe variations in BEC in a subgroup of patients with consecutive measurements, with 17% of the patients in the low BEC group having BEC >300 in the later assessment. How were those patients finally classified? I suggest either considering those patients with elevated BEC at exacerbation as « T2high » (similarly to severe asthma), or excluding them. I also suggest that BEC in the two years before enrollment is considered to re-categorise the subjects accordingly.

Response: All patients were set up on LT-NIV at least 4 weeks after their last exacerbation. We have further clarified this in 2.1. Design and subjects. Some patients took systemic steroids for their autoimmune/rheumatic disease, the prevalence of these subjects was not different between the two groups (please see Table 1). We used the first BEC test when categorising the patients into the two groups. We have now explained this in the Discussion, 5th paragraph. We have already excluded patients with elevated BEC at exacerbation. Historical BEC data was not available. This has been added to the limitations. Please, see Discussion, last paragraph.

Comment: The mean duration of follow-up is not provided. Given the date of LT-NIV between 2011 and 2023, it should be highly heterogeneous between patients. I therefore wonder if the survival curves indicating « 100 months follow-up » in the legend of the x-axis is correct. The number of patients followed up for 20, 40, … months should at least appear under the x-axis.

Response: Thank you very much. We provided the requested results. Instead of adding them to the figures, we have summarised them in a separate table. Please, see Table 4.

Comment: The title should does not totally reflect the results and should be toned-down

Response: Thank you we have modified the title as suggested.

Comment: Table 1 :

BEC ranges should be provided for both groups

AHR could be added

Response: BEC were provided as requested. Airway hyperresponsiveness is not routinely performed in patients with COPD; therefore, unfortunately, this data is not available.

Reviewer 2 Report

Comments and Suggestions for Authors

Overall, this manuscript tackles a clinically pertinent and underexplored question: whether the eosinophilic phenotype predicts better response to LT-NIV in hypercapnic COPD; and draws on a reasonably large cohort with prolonged follow-up, yielding a consistent signal of benefit after multivariable adjustment. However, the retrospective single-center design, classification based on a single blood eosinophil count despite notable variability, and some inconsistencies between the Abstract and Results temper the strength of the inferences. The clinical interpretation would be strengthened by modeling person-time rates (e.g., Poisson/negative binomial), treating eosinophils as a continuous predictor with sensitivity analyses across thresholds, documenting therapy changes, effective NIV adherence, and, if available, causes of death. Specific comments:

Major

  1. Align Abstract and Results. Please clarify whether the survival p=0.01 in the Abstract refers to adjusted Cox rather than the unadjusted log-rank. Use consistent tests across different sections and avoid labeling an adjusted result as “log-rank.”
  2. Address eosinophil count variability. Given 19% category switching and only 11% persistent high eosinophil count, it would be recommendable to add sensitivity analyses.
  3. Account for NIV history (adherence/effectiveness). Beyond mean nightly hours, consider including time-on-device as a time-varying covariate or stratify by adherence (e.g., ≥4 h/night) to mitigate compliance confounding.
  4. Therapy changes during follow-up. Specify ICS/LABA/LAMA treatments and, if feasible, adjust for these or provide sensitivity analyses, given ICS–eosinophil interactions.
  5. Mortality details. If retrievable, provide causes of death.

Minor

  1. Kaplan–Meier figures: add numbers-at-risk and include HR with 95% CI and the exact test used in figure legends.
  2. Table 1 enhancements: add “frequent exacerbator” status (either ≥2/year or ≥3/year) to facilitate comparisons.

Comments on the Quality of English Language

The English used in the manuscript is clear and understandable, with an appropriate scientific register, but it needs editorial polishing to sound more natural and consistent to native English-speaking readers.

Author Response

Comment: The clinical interpretation would be strengthened by modeling person-time rates (e.g., Poisson/negative binomial), treating eosinophils as a continuous predictor with sensitivity analyses across thresholds, documenting therapy changes, effective NIV adherence, and, if available, causes of death.

Response: Thank you for your comments. We agree that in prospective studies, modelling person-time rates could strengthen the interpretation. However, due to the retrospective nature of the study, different enrolment and follow up times, we chose Cox-regression analysis. Following your comments, further sensitivity analyses were performed by treating BEC as a continuous predictor and using different cut offs. See Methods. Following these, only a cut off of BEC>300 cells/µL was found to be significant. See Results and Discussion, 3rd paragraph. Please, find our answers below for the further comments.

Comment: Align Abstract and Results. Please clarify whether the survival p=0.01 in the Abstract refers to adjusted Cox rather than the unadjusted log-rank. Use consistent tests across different sections and avoid labeling an adjusted result as “log-rank.”

Response: Thank you for the suggestion. We have added the adjusted results to the abstract and it is now corrected. Following adjustment on the hours of NIV use, the p value changed to 0.02.

Comment: Address eosinophil count variability. Given 19% category switching and only 11% persistent high eosinophil count, it would be recommendable to add sensitivity analyses.

Response: We fully agree with the reviewer’s comment; however, due to the low numbers sensitivity analyses were not performed. We have expanded the Discussion, paragraphs 5th and 7th to explain this.

Comment: Account for NIV history (adherence/effectiveness). Beyond mean nightly hours, consider including time-on-device as a time-varying covariate or stratify by adherence (e.g., ≥4 h/night) to mitigate compliance confounding.

Response: Following the reviewer’s comment, the analyses were adjusted for average hours of NIV use. We used hours of usage as a continuous covariate as some patients require prolonged hours of ventilation, whilst in others minimal hours are enough to correct their hypercapnia. Please, see the revised Abstract, Methods and Results.

Comment: Therapy changes during follow-up. Specify ICS/LABA/LAMA treatments and, if feasible, adjust for these or provide sensitivity analyses, given ICS–eosinophil interactions.

Response: Following the reviewer’s comment we have provided detailed breakdown of patients taking ICS/LABA/LAMA. We have provided sensitivity analyses on patients taking triple therapy and the results were similar to the overall population. Due to the low number of subjects, no analyses were performed in patients taking specific combinations or patients not on triple therapy. Please, see Methods, Results and Discussion 6th paragraph. The analyses were not adjusted on treatment modifications. This has already been listed as one of the limitations in the manuscript. Please, see Discussion, 7th paragraph.

Comment: Mortality details. If retrievable, provide causes of death.

Response: Unfortunately, due to the nature of the study, whilst we were able to assess the fact of death, causes of mortality were not available. Please, see Discussion, 3rd paragraph.

Comment: Kaplan–Meier figures: add numbers-at-risk and include HR with 95% CI and the exact test used in figure legends.

Response: We have added risk ratios with 95% CI to the figure legends.

Comment: Table 1 enhancements: add “frequent exacerbator” status (either ≥2/year or ≥3/year) to facilitate comparisons.

Response: We provided the requested information.

Comment: The English used in the manuscript is clear and understandable, with an appropriate scientific register, but it needs editorial polishing to sound more natural and consistent to native English-speaking readers.

Response: The manuscript has been double checked by an English native speaker.

Reviewer 3 Report

Comments and Suggestions for Authors

1. The authors have undertaken a retrospective cross-sectional study of patients with severe COPD, chronic and stable hypoxaemia and CO2 retention who were commenced on long term non invasive ventilation (LT-NIV). They divided the population into those who had, at least on one occasion at baseline, an  elevated blood blood eosinophil count, with a high cutoff of 300 cells per microliter. The total number involved was relatively small although respectable with 163 followed up for 12 months, and a much smaller number of 37 in the high eosinophil group. In the event, this latter group of interest had a better survival over 12 months then the control group although it had been more un-stable in the 12 months previous. There was a small improvement in the exacerbation rate in this group post initiation of an NIV, but the most striking difference was in the totality and (non-specified) hospital admission.

 2. The study is quite interesting and introduces some novel concepts, at least to me. The hypothesis and rationale for the study as given in the introduction is fairly weak, and I suggest that this needs beefing up with perhaps more emphasis on the theory that emerges in discussion on an interaction between hyper-eosinic airway inflammation and hyper-carbia. If there is not biological plausibility to start with then it is difficult to know whether the study was worth doing and whether the outcomes could just be random change or related to regression towards the mean in the worst clinically-active group pretreatment.

3. There are some other factors that I think could and indeed should be included in the discussion as potential confounders: No data is given on any history of asthma especially in the high eosinophil group or differential pack years of smoking rather than a dichotomous division into (ex)-smokers or not. I presume that the authors just do not have this information. Similarly, is it possible that the attention given to patients at the time of starting NIV may have improved compliance with ICS in the worse clinical group who were unstable and had a high eosinophil count; again there was no follow up of a eosinophil count to know whether this dropped as a result or better compliance. There is probably no way around these issues given the nature of the study and the data capture, but a fuller disclosure of the inadequacies is important especially what needs to be teased out in follow-up of these interesting and intriguing findings. The already volunteer some important aspects such as finding out what the extent excess deaths were due to in the control group, or the converse what these individuals of interest.did not die of!

4. I am not sure that it is worth dividing acute exacerbations into infective or noninfective as in general it always strikes me as being somewhat arbitrary. It is not a concept or definition that appears in the methods (or indeed anywhere in the paper). I suggest amalgamation of these events.

5. To the credit of the authors, they do not claim that this is much more than an exploratory investigation, but perhaps this should be emphasised rather more in Intro and Discussion. 

Author Response

Comment: 1. The authors have undertaken a retrospective cross-sectional study of patients with severe COPD, chronic and stable hypoxaemia and CO2 retention who were commenced on long term non invasive ventilation (LT-NIV). They divided the population into those who had, at least on one occasion at baseline, an  elevated blood blood eosinophil count, with a high cutoff of 300 cells per microliter. The total number involved was relatively small although respectable with 163 followed up for 12 months, and a much smaller number of 37 in the high eosinophil group. In the event, this latter group of interest had a better survival over 12 months then the control group although it had been more un-stable in the 12 months previous. There was a small improvement in the exacerbation rate in this group post initiation of an NIV, but the most striking difference was in the totality and (non-specified) hospital admission.

Response: Thank you for the supportive comment.

Comment:  2. The study is quite interesting and introduces some novel concepts, at least to me. The hypothesis and rationale for the study as given in the introduction is fairly weak, and I suggest that this needs beefing up with perhaps more emphasis on the theory that emerges in discussion on an interaction between hyper-eosinic airway inflammation and hyper-carbia. If there is not biological plausibility to start with then it is difficult to know whether the study was worth doing and whether the outcomes could just be random change or related to regression towards the mean in the worst clinically-active group pretreatment.

Response: Thank you very much. This is now better articulated in the revised manuscript. Please, see Introduction.

Comment: 3. There are some other factors that I think could and indeed should be included in the discussion as potential confounders: No data is given on any history of asthma especially in the high eosinophil group or differential pack years of smoking rather than a dichotomous division into (ex)-smokers or not. I presume that the authors just do not have this information. Similarly, is it possible that the attention given to patients at the time of starting NIV may have improved compliance with ICS in the worse clinical group who were unstable and had a high eosinophil count; again there was no follow up of a eosinophil count to know whether this dropped as a result or better compliance. There is probably no way around these issues given the nature of the study and the data capture, but a fuller disclosure of the inadequacies is important especially what needs to be teased out in follow-up of these interesting and intriguing findings. The already volunteer some important aspects such as finding out what the extent excess deaths were due to in the control group, or the converse what these individuals of interest.did not die of!

Response: In the revised manuscript we added data on comorbidities, including asthma, allergic rhinitis and eczema. Data on smoking was already available. We fully agree with your plausible explanation that patients became more adherent to inhalers once they were under our care. This effect might be more significant in those with high BEC. We have added this explanation to the manuscript. See Discussion, 6th paragraph. Unfortunately, specific causes on death were not available, but we have expanded the Discussion, 3rd paragraph to discuss this in more detail. 

Comment: 4. I am not sure that it is worth dividing acute exacerbations into infective or noninfective as in general it always strikes me as being somewhat arbitrary. It is not a concept or definition that appears in the methods (or indeed anywhere in the paper). I suggest amalgamation of these events.

Response: We decided to keep infective exacerbations in the manuscript as they can shed light into the mechanism of exacerbation reduction in the high BEC group. We expanded the Methods and Discussion, 4th paragraph.

Comment: 5. To the credit of the authors, they do not claim that this is much more than an exploratory investigation, but perhaps this should be emphasised rather more in Intro and Discussion. 

Response: We expanded the Discussion emphasising further limitations of the project. The exploratory nature of the study is described in Methods.

Round 2

Reviewer 1 Report

Comments and Suggestions for Authors

All my comments have been adressed.

Author Response

Thank you very much

Reviewer 3 Report

Comments and Suggestions for Authors

The authors should be congratulated with the way they have industriously modified their manuscript, which I think improves it substantially. This is interesting work which will alert the respiratory community to an intriguing phenomenon, which now requires as always more research. 

Author Response

Thank you very much